# Does the Association between Guardians’ Sense of Coherence and their Children’s Untreated Caries Differ According to Socioeconomic Status?

**DOI:** 10.3390/ijerph17051619

**Published:** 2020-03-03

**Authors:** Akiko Mizuta, Jun Aida, Mieko Nakamura, Toshiyuki Ojima

**Affiliations:** 1Department of Community Health Nursing, Hamamatsu University School of Medicine, 1-20-1, Handayama, Higashi-ku, Hamamatsu, Shizuoka 431–3192, Japan; 2Department of International and Community Oral Health, Tohoku University Graduate School of Dentistry, 4-1 Seiryouchiyou, Aobaku, Sendai, Miyagi 980-8575, Japan; j-aida@umin.ac.jp; 3Department of Community Health and Preventive Medicine, Hamamatsu University School of Medicine, 1-20-1, Handayama, Higashi-ku, Hamamatsu, Shizuoka 431–3192, Japan; miekons@hama-med.ac.jp (M.N.); ojima@hama-med.ac.jp (T.O.)

**Keywords:** dental caries, sense of coherence, economic status

## Abstract

Untreated caries is the most prevalent disease in the world. A sense of coherence (SOC) is believed to contribute to oral health. We aimed to clarify the association between guardians’ SOC and their children’s caries based on socioeconomic status (SES) in Japan. This study’s subjects were Japanese public junior high schoolers (N = 1730), aged 12–15, and their guardians in Kosai City. We administered a questionnaire survey among guardians in 2016 to assess their SOC and family environment. With their students’ consent, public junior high schools shared the results of the dental examinations that were part of their school physicals. Multivariate logistic regression was conducted to clarify the association between guardians’ SOC and their children’s untreated decayed permanent teeth. We also conducted a stratified analysis according to a relative poverty line. We observed in the multivariate regression a significant inverse association between children’s untreated decay and their guardians’ SOC (OR 0.93, 95%CI 0.87–1.00). The association of SOC was stronger in the low economic group (OR 0.64, 95%CI 0.43–0.95). Guardians with higher SOC were associated with children having fewer caries. Guardians’ SOC is a factor for the prevalence of caries and access to dental care, especially among children with low economic status.

## 1. Introduction

Untreated caries in permanent teeth account for 35% of the global occurrence of disease for all ages combined, making tooth decay the most prevalent ailment in the world [1]. Oral diseases impair communication, emotional expression, and food intake, and restrict participation in activities at social events, school, work, and home [2]. Among children with low socioeconomic economic status (SES), exposure to infections, smoking, and poor oral hygiene habits cause periodontal disease [3]. As a result, tooth loss occurs [4].

Per person, the average DMFT (Decayed, Missing, and Filled Teeth) index was 1.86 among 12-year-olds worldwide in 2015. In comparison with the global value of DMFT, Japan had a low value (0.2) in 2016 [5]. However, the DMFT index increased to 3.1 for 15–24 years as age increased. The proportion of caries increased rapidly from 10.3% among 12-year-olds to 44.4% among 13-year-olds [6]. Hence, adolescence is an important period for oral health education.

Antonovsky defined a sense of coherence (SOC) as an individual’s salutogenic resources and explained it as an individual’s generalized emotional-cognitive perception of and relative control over the stimuli bombarding him [7]. The SOC scale has been used in many countries to confirm the validity and reliability of strategies for managing stressful situations and promoting health [8]. An association between mothers’ higher SOC and lower tooth decay has been found among preschool children [9] and adolescents [10].

One survey among school children showed that the amount of caries differs significantly according to SES at the community level [11]. A few studies have found an association between caries and household income among preschool children [12,13,14]. However, improvements in economic status from childhood to adolescence have been shown to contribute only to increased dentist consultations and are not associated with healthy behaviors, such as more frequent teeth brushing or lower sugar consumption [15]. Previous studies have reported that a mother’s schooling and employment status are associated with impacting oral hygiene performance in terms of physical, psychological, and social dimensions of daily living [16]. In Japan, where universal healthcare insurance covers dental treatment, absolute inequalities and lower parental education increase the rate of treatment for caries [17]. In particular, a mother’s low education level is a significant risk factor for caries [12,13,14,18,19]. Health inequalities expand in relation to access to dental treatment in children.

The positive association of SOC and oral health-related behavior has been reported among adults [20] and adolescents [21]. It is necessary to explore the possibility of parental psychological interventions to promote dental health among adolescents, who depend on guardians to access to dental care. However, to the best of our knowledge, no studies have investigated the association between guardians’ SOC and caries among adolescents with low SES. We hypothesized that guardians with low SES who have high SOC would be associated with children’s good oral health. Therefore, we aimed to clarify the association between guardians’ SOC and the untreated decay in their children’s permanent teeth, according to SES.

## 2. Materials and Methods

### 2.1. Study Subjects

This study’s subjects were Japanese students (N = 1730) from the five public junior high schools of Kosai City, Shizuoka Prefecture, and their guardians. Japanese junior high school students range in age from 12 to 15 years. An anonymous questionnaire survey was administered in June 2016. Answering the anonymous self-assessment questionnaire was considered to indicate consent to participate. Participants were provided the opportunity to decline. Blank questionnaires of students and their guardians who rejected participation in this survey were also collected. A check box was provided on the parent’s questionnaire to indicate consent to participate. Results of students’ dental examinations and questionnaires were linked with their parents’ questionnaires using symbols. Guardians (N = 1370) filled out and returned the questionnaire. With students’ consent, the schools shared the results of the dental examinations that were part of the school physicals. We included data from the 1056 respondents (61.0%) who answered questions regarding income and SOC.

### 2.2. Measurement

During the annual schoolwide checkup conducted from April to June 2016, the school dentist diagnosed untreated decay in permanent teeth by means of a dental mirror. The examinations were performed in a schoolroom where the lighting was set to the prescribed standard. A tooth affected by any of the stigmata of caries is designated as “DMF” (decayed, missing, or filled) [22]. We focused on untreated decayed permanent teeth, because adolescents depend on their guardians to access dental care. We dichotomized untreated decayed permanent teeth into two categories (0 or ≥1), following the method of previous studies [9].

A self-administered questionnaire was distributed to guardians at five public junior high schools. It included items on family environment, such as yearly household income, SOC, educational attainment, marital status, and age. A three-item SOC questionnaire was developed to assess a three-component construct based on Antonovsky’s original 29-item instrument: comprehensibility, manageability, and meaningfulness [23,24,25,26]. It has been used for large-scale surveys. We assessed the participating guardians’ SOC using the short, three-item version of the scale developed by University of Tokyo Health Sociology (SOC-3-UTHS), which has high internal consistency (Cronbachα = 0.84) and whose levels of convergent and concurrent validities have been indicated (*r* = 0.51) [26]. There were seven response options for the questions, ranging from “not at all applicable” to “very applicable.” The total score range was 3 to 21. A higher score meant a higher SOC.

We asked economic status of 12 stages; less than 1 million yen, 1 million or more to less than 2 million yen divided into 2 stages by 500,000 yen unit, 2 million yen or more to less than 10 million yen divided into 8 stages by 1 million yen unit, and 10 million yen or more (approximately less than 9090 to 90,909 dollars or over). Annual equivalent income was established by computing the median of each stage and dividing it by the square root of household members. The unit of annual equivalent income was set at 10 million yen. Then, we divided it into two categories “low economic status” or “high economic status” (<1.32 million yen or ≥1.32 million yen), using a poverty line that defines poverty as half of the median of annual equivalent income, according to a national survey of family income and expenditure [27].

There were five response options for the question on educational attainment: elementary school, junior high school, high school, junior college or vocational-technical school, and university or higher. Hence, educational attainment was classified into four categories, following previous research [17]. Parental education up to high school level or lower was categorized as lower educational attainment, while education up to college level or higher was categorized as higher educational attainment (< 18 or ≥ 19 years). There were three response options for marital status: “married”, “single parent”, and “remarried”.

### 2.3. Statistical Analyses

We distributed the frequency for each variable. First, to evaluate the influence of family environment on the number of caries in a child, we examined the association between covariates and decayed permanent teeth among junior high school students using univariate logistic regression analysis. Next, we conducted multivariate logistic regression using a child’s decayed permanent teeth as an objective variable and their guardians’ SOC as the explanatory variable, adjusting for equivalent annual income, educational attainment, marital status, age, and the student’s gender and age. Furthermore, we conducted a stratified analysis based on relative poverty to compare the odds ratios (ORs) of guardians’ SOC between students with low SES and those with high SES. The interaction between equivalent annual income and educational attainment and guardians’ SOC was evaluated by including the interaction terms in multivariate regression, respectively. For all analyses, alpha was 0.05. We used STATA version 14.0 (STATA Corp LP., College Station, TX, USA) for statistical analysis.

### 2.4. Ethical Statement

All subjects gave informed consent for inclusion before they participated in the study. The study was conducted per the Declaration of Helsinki, and the protocol was approved by the Ethics Committee of Hamamatsu University School of Medicine (E15-293).

## 3. Results

### 3.1. Distribution of Participant

Among the guardians, most of the respondents were mothers (88.2%); 11.5% were fathers, and 0.3% were grandmothers. Descriptive statistics of the study subjects are shown in Table 1. Per person, the average DMFT index was 0.6 (standard deviation (SD) 1.2) for 12–13 years, 0.6 (SD 1.3) for 14 years, and 0.8 (SD 1.7) for 15 years. The proportion of decayed teeth among adolescents was 9.37% for 12–13 years, 9.02% for 14 years, and 11.78% for 15 years. Guardians’ SOC was 15.1 (SD 3.2, range 3–21). The details of SOC were as follows: 15.7 (SD 2.9) for fathers, 15.0 (SD 3.3) for mothers, and 16.7 (SD 4.2) grandmothers. The relative poverty rate was 7.8%, that is, an equivalent annual income of less than 1.32 million yen, according to the poverty line. Households with both parents having high educational attainment comprised 31.4% of the respondents, and households with a biological parent present comprised 86.1%.

### 3.2. Family Environment and Children’s Caries

Table 2 shows crude and adjusted odds ratios (ORs) and 95% confidence intervals (CIs) from the estimates of the applied logistic regression. In univariate regression, equivalent annual income (unit 10 million), guardians’ SOC, and educational attainment were significantly associated with decayed teeth (OR 0.12, 95%CI 0.02–0.67; OR 0.93, 95%CI 0.93–0.87; and OR 0.48, 95%CI 0.28–0.83, respectively). In multivariate regression, a significant inverse association was observed between guardians’ SOC, educational attainment, and decayed teeth (OR 0.93, 95%CI 0.87–1.00; and OR 0.51, 95%CI 0.28–0.94, respectively). Equivalent annual income had a significant marginal association with decayed teeth (OR 0.14, 95%CI 0.02–1.04).

### 3.3. Guardians’ Sense of Coherence and their Children’s Caries among the Low Economic Group

Table 3 shows the results of multivariate logistic regression according to the relative poverty line. The association between guardians’ SOC and decayed teeth was stronger among the low economic group (OR 0.64, 95%CI 0.43–0.95) than the high economic group, although the interaction terms of equivalent annual income and guardians’ SOC were not statistically significant. Students with two parents with high educational attainment comprised 13.6% (*n* = 9) in the low economic group and 32.6% (*n* = 308) in the high economic group. In the high economic group, educational attainment was significantly associated with decayed teeth (OR 0.50, 95%CI 0.27–0.91). The interaction terms of educational attainment and guardians’ SOC was not significant.

## 4. Discussion

Through this study, we have found that guardians’ higher SOC was associated with their children having fewer caries. This result suggests that guardians’ SOC is a psychosocial determinant of caries and salutogenic resources of a child’s oral health. There may be an interaction between guardians’ SOC and economic status on their children’s caries. To the best of our knowledge, the present study is the first in Japan to clarify the association between guardians’ SOC and their children’s caries and observe the modification effect of economic status on guardians’ SOC.

The significant association between guardians’ SOC and their children’s caries was observed after we adjusted for economic status. Bonanato et al. reported in preschool children that mothers’ SOC was associated with their children’s caries [9]. Our result is consistent with Freire et al., who observed the association in adolescence [10]. SOC is not only useful for promoting oral health in adults [25,26] but also an important factor for children. Previous studies investigated the association of mothers’ SOC with children’s caries; they did not clarify the effect of economic status on the relationship. Our study found a significant association between guardians’ higher SOC and children having fewer caries among students with low economic status below the poverty line. SOC is the central hypothesis behind the model for coping with psychosocial stressors [7]. A mother’s coping style that includes a higher probability of obtaining information and using problem-focused management and a lower probability of evading her responsibility is associated with her children having good oral health care [28]. Poor SOC was more common among the lower-income group than in the higher-income group [29]. However, we believe enhancing guardians’ SOC might be more useful for managing adverse situations than promoting health at the level of high economic status. Adolescents whose mothers had higher SOC were less likely to attend the dentist when having trouble than those whose mothers had lower SOC [10]. In contrast, adolescents whose mothers had higher SOC were more likely to utilize dental care services and a dentist for check-ups [30]. Guardians’ SOC is an important psychological factor in determining access to dental care and thus addressing the social inequalities of children’s oral health. Adolescents depend on guardians to access oral care.

Higher economic status was strongly associated with fewer caries in the present study. Edelstein reported that Medicaid-eligible children have twice the numbers of caries and visits for pain relief, compared to children with higher household incomes in the United States [31]. Aida et al. suggested that the rate of caries treatment is higher for preschool children with lower SES in Japan, with a significant widening of absolute inequalities along with the growth of the children [17]. Even though universal health insurance covers dental care in Japan, economic status remains a determinant of oral health in children.

Compared to parents who had low educational attainment, those with high educational attainment were associated with children having a low caries count. Our study demonstrated the association might continue between guardians’ educational attainment and children’s caries after adjusting for economic status. Previous studies have reported findings on the association between parental education and children’s caries [12,13]. Oral health disparity caused by education continues into adolescence [14,15,16,17,32]. Parental education is one of the main determinants of caries among children. Although low SOC was more common among the low-educated parents than the highly educated parents [29], our study did not show an interaction between educational attainment and guardians’ SOC on caries. However, we observed a significant association between guardians’ higher educational attainment and low caries counts in children with higher economic status. SOC emphasizes health promotion as a salutogenic approach [7]. In our study, SOC might be effective as a promoter of dental health only in cases of low economic status. Parents’ oral health-related attitudes and behaviors were significantly associated with their children’s oral health-related attitudes [33] and behaviors, such as brushing their teeth twice a day [34]. Furthermore, mothers’ untreated caries and tooth loss were associated with their children’s caries experiences. An educational program that teaches health-promoting behaviors in terms of dental caries should be provided to parents.

Our study has a few limitations. First, as a cross-sectional study, this research limited the vocational-technical establishment of the causal relationship of guardians’ SOC and their children’s caries. Second, although our obtaining of familial information was a strength, some information might not have been accurately reported due to recall bias. Third, our study collected data from five schools, all of which were located in one city. Hence, we are unable to generalize our results as they may differ from general findings across Japan. Our study has several strengths. We inquired directly about annual household incomes from guardians through this study and gathered the anthropometric data of adolescents that were diagnosed by a school dentist. However, poverty may have been underestimated, because parents’ questionnaires were collected by school personnel. The number of dentists per 100,000 population per province (62.9) is smaller than the national number (80.0) [35]. The DMFT index in our study population (7.0) is lower than the mean value of the national population (0.83), which included subjects aged between 12 and 15 in 2016 [36]. Further large-scale studies that include these factors are needed to confirm the association.

## 5. Conclusions

Guardians’ higher SOC was associated with children’s lower caries. Among children with low economic status, guardians’ SOC is an important salutogenic resource.

## Figures and Tables

**Table 1 ijerph-17-01619-t001:** Descriptive statistics for guardians and their children.

Variables	Contents	n/Mean	%/SD
Guardian			
	Father	121	11.5
	Mother	929	88.2
	Grandmother	3	0.3
Guardians’ SOC		15.1	3.2
	Father	15.7	2.9
	Mother	15.0	3.3
	Grandmothers	16.7	4.2
Equivalent Annual Income	< 1.32 million yen (poverty line ^a^)	82	7.8
Educational Attainment	Low for both parent (˂ college)	367	36.4
	High for mother and low for father	159	15.8
	High for father and low for mother	166	16.5
	High for both parents (≥ college)	317	31.4
Marital Status	Married	908	86.1
	Single parent	117	11.1
	Remarried	30	2.8
Age	Father	45	5.5
	Mother	43	4.7
Students’ DMFT			
	12–13 years	0.6	1.2
	14 years	0.6	1.3
	15 years	0.8	1.7
DT	≥1	106	10.0
MT	≥1	14	1.3
FT	≥ 1	237	22.4
Gender	Boys	544	51.5
	Girls	512	48.5
School Grade	7th grade	331	31.3
	8th grade	377	35.7
	9th grade	348	33.0

Note. Sample size is different due to missing values. SOC: sense of coherence. DMFT: decayed, missing, and filled teeth. DT: decayed teeth. MT: missing teeth. FT: filled teeth. ^a^ Half of the median of annual equivalent income.

**Table 2 ijerph-17-01619-t002:** Odds ratios for SOC, income, education, age, and interaction between SOC and income in relation to a child’s decayed permanent teeth ^b^.

		Univariate					Multivariate ^a^ (N = 957)				Interaction
		N	OR	95% CI		*p*	OR	95% CI		*p*	*p*
Guardian’s SOC		1056	0.93	0.87	0.99	0.017	0.93	0.87	1.00	0.041	0.031
Equivalent Annual Income	12 stages (unit: 10 million yen ^c^)	1056	0.12	0.02	0.67	0.015	0.14	0.02	1.04	0.055	0.085
Educational Attainment	ref: Low for both parent (˂ college)	1009					1				
	High for mother and low for father		0.92	0.52	1.63	0.785	1.03	0.57	1.86	0.933	0.011
	High for father and low for mother		0.73	0.40	1.32	0.296	0.82	0.44	1.52	0.526	0.461
	High for both parents (≥ college)		0.48	0.28	0.83	0.008	0.51	0.28	0.94	0.030	0.963
Marital Status	Married	1055					1				
	Single parent		1.14	0.61	2.10	0.685	1.87	0.78	4.48	0.160	0.054
	Remarried		1.01	0.30	3.40	0.987	0.62	0.14	2.76	0.534	0.477
Age	Father	982	1.02	0.98	1.06	0.351	1.03	0.99	1.08	0.175	0.239
	Mother	1027	0.99	0.95	1.03	0.626	0.98	0.92	1.04	0.497	0.348
Interaction term SOC × Equivalent annual income											0.129

Note. SOC: sense of coherence. OR: odds ratio. ^a^ Adjusted for income, education, marriage, age, and child’s gender and school grade. ^b^ Decayed permanent teeth of 0 or ≥ 1. ^c^ Equivalent annual income as a quantitative variable given in units of 10 million yen.

**Table 3 ijerph-17-01619-t003:** Odds ratios for SOC on decayed permanent teeth stratified by economic status.

		Low (< 1.32 million yen, N = 44)				High (≥1.32 million yen, N = 903)			
		OR	95% CI		*p*	OR	95% CI		*p*
Guardians’ SOC		0.64	0.43	0.95	0.027	0.95	0.88	1.02	0.126
Educational Attainment	ref: Low for both parent (˂ college)	1				1			
	High for mother and low for father	0.28	0.01	6.28	0.425	1.01	0.55	1.86	0.979
	High for father and low for mother	0.40	0.02	7.75	0.541	0.85	0.45	1.61	0.621
	High for both parents (≥ college)	-				0.50	0.27	0.91	0.025
Marital Status	Married	1				1			
	Single parent	7.47	0.55	102.00	0.132	1.45	0.41	5.07	0.561
	Remarried	-				0.77	0.17	3.42	0.732
Age	Father	0.88	0.68	1.15	0.348	1.03	0.98	1.09	0.175
	Mother	1.06	0.79	1.41	0.705	0.97	0.91	1.03	0.301

Note: SOC: sense of coherence. OR: odds ratio. Adjusted education, marriage, age, and child’s gender and school grade. ^a^ Equivalent annual income as a qualitative variable divided by poverty line.

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
