# Peer review of "Does the Association between Guardians’ Sense of Coherence and their Children’s Untreated Caries Differ According to Socioeconomic Status?"

_ijerph, 2020, doi:10.3390/ijerph17051619_

Round 1
Reviewer 1 Report
The present study aimed to clarify the association between guardians’ Sense of Coherence and their children’s caries based on socioeconomic status (SES) in a city of Japan. The manuscript is well written although some informations about methods are missed.
Abstract
The authors should add the city of Japan where this study was undertaken.
2.1. Study subjects
“...with students’ consent...”. The consent must be declared by students and AUTHORIZED by the guardians as they are under age. Also, no statement was presented about the consent from guardians questionnaire. Was the same questionnaire used to students and guardians? Please, add these informations within the text.
Was the data collection made by 1 or more interviewers? Were they previously trained?
2.2. Measurement
Line 81 – Was the dentist calibrated before examination of the oral cavity? The authors shall insert more specifications about the exam: Was it in a dental office? Using lights? Mirrors? How long have the examinations taken (months, days…?)
Line 84 – “We dichotomized untreated decayed permanent 84 teeth into two categories (<0 or ≥1)”. Is there less than 0 in this classification?
Lines 85 and 86 – “following the method of previous studies that reported on the 85 relationship between mothers’ SOC and decayed teeth in preschool children [9].” Please, correct if there were more than 1 study.
Line 97 – In my opinion the authors should convert categories in dollars. It will be easier to understand the situation.
Line 106 - Previous studies reported differences in the association between the educational attainment of 106 mothers and parents and their children’s caries [12, 13]. Please, this statement should be at the discussion section.
Results
In table 3 – I could not find within the table letter ”a” mentioned below.
Discussion
Line 211 – “In our study, SOC might be effective as a promoter of dental health only in stable economic environments.” Did the authors mean as stable economic environments high economic status? If not, please rewrite this sentence.
Line 221 – “Third, our study collected data from one city in Japan.” It was only 1 school in this city. Could you extrapolate to the whole city? So, in my opinion the authors should state data presenting the homogeneity within the regions of this city.
Line 225 – The authors stated “The DMFT index in our study population was higher than that of the national 225 survey.”. Please, add a short explanation for what reasons the authors think this happened.
Conclusion
The authors should remove from “and is associated with……” until the end of conclusions.
Reviewer 2 Report
The manuscript deals with the association between Guardians’ sense of coherence and their children’s untreated decay and how it differ according to socioeconomic status.
In general this manuscript has merit and deserves futher attention. Unfortunatley the presentation of the data is not optimal and should be improved, maybe a figure would be benefitial as well.
Find below some more aspects to consider:
In general, you mention the term “cavity” frequently; please check and adapt the adequate wording - whether you really refer to “cavitated carious lesions”, or talk about “caries experience”, “caries levels “or “untreated dental decay”, as they are all slightly different from another. Most likely you refer to the D component in the DMFT, right?
Please be precise.
abstract:
l 20. Change to children’s caries experience …
mention also the sample N for children / guardians
l. 30 as this is a cross-sectional study, I dont think it is possible to referr to incidence, rather risk...
Chapter 2.1.:
How did you match the child questionnaire with the dental status?
The way I understood your methods, it cannot be an anonymous questionnaire survey in my opinion...as you related the answer of the parents to the DMFT of each child.
Tab. 1
In general the labelling is very confusing with n / mean -> reorganize this table and its data
mean for father and mother, grandmother? This is not clear to me. N of total sample would be good D M F components of students are not clear Students DMFT according to age (12yr -> x DMFT ; 13 yr -> xDMFT and so on) would be be good to mention here as well (as in the text l. 133ff)Tab. 2 and 3
please also mention here how you categorized DT in the children, all abbreviations need to be explained, not clear on first sight what is meant with|
Low (N = 44) |
High (N = 903) |
Discussion
I am not sure whether this term is completely appropriate “population-based study”. To what extent to you think your sample is representative for Japanese adolescents/families? What kind of children attend the selected schools?
l. 186 - I agree: “However, we believe enhancing guardians’ SOC might be more 186 useful for managing adverse situations than promoting health at the level of high economic status.”
l. 209 How do you explain the correlation of low SES and low SOC, and why do some despite of low SES a higher SOC?
L. 225 I miss a reference for this statement: “The DMFT index in our study population was higher than that of the national survey.”
Conclusion:
Please rephrase the conclusion
Especially the sentence “ Guardians’ higher SOC was associated with children’s lower cavity count.” has a very uncommon wording.
Moreso, I don’t think the aspects of fluorides should be part of the conclusion, as it does not relate to the study.
Reference 19: why capital letters?
Reviewer 3 Report
This study investigated the relationship between guardian’s SOC, children’s untreated caries, and socioeconomic status among five middle school students in Japan. The authors obtained valuable information on socioeconomic status such as parental academic background and annual income. The authors clarified that guardian’s SOC was inversely associated with children’s untreated decayed teeth, and that the association was stronger in the low economic group. The results were reasonable, however, there are some concerns in methodology that I listed below.
Comments
Overall in manuscript
1
The authors categorized annual household incomes into 12 groups, from 1 million to ten million yen. However, they set 10 million yen as an independent variable in Table 2. I cannot understand the reason. The authors should replace it to 1 million yen or remain 12(or 3 to 5) categories.
2 The interaction between guardian’s SOC and annual income was not significant in Table 2 (P=0.129). However, the authors conducted stratified analyses by income level. In addition, the number of participants in low economic status was only 44 (in Table 3). I wonder if the analysis conveyed a robust association.
Round 2
Reviewer 3 Report
I still have one concern about the result in Table 2. The equivalent annual income had a strong relationship with dental caries (OR=0.14, 0.02-1.04) but it is not statistically significant. I wonder if the independent variable has fully variance and if the authors really used 12 stages as continuous variance.